# Epigenetic Biomarkers in Colorectal Cancer Patients Receiving Adjuvant or Neoadjuvant Therapy: A Systematic Review of Epidemiological Studies

**DOI:** 10.3390/ijms20153842

**Published:** 2019-08-06

**Authors:** Martina Barchitta, Andrea Maugeri, Giovanni Li Destri, Guido Basile, Antonella Agodi

**Affiliations:** 1Department of Medical and Surgical Sciences and Advanced Technologies “GF Ingrassia”, University of Catania, via S. Sofia, 87, 95123 Catania, Italy; 2Department of General Surgery and Medical-Surgical Specialties, University of Catania, via S. Sofia, 78, 95123 Catania, Italy

**Keywords:** DNA methylation, miRNA, prevention, Public Health, Epidemiology, colon cancer, rectal cancer, CIMP

## Abstract

Colorectal cancer (CRC) represents the third-most common cancer worldwide and one of the main challenges for public health. Despite great strides in the application of neoadjuvant and adjuvant therapies for rectal and colon cancer patients, each of these treatments is still associated with certain adverse effects and different response rates. Thus, there is an urgent need for identifying novel potential biomarkers that might guide personalized treatments for specific subgroups of patients. However, until now, there are no biomarkers to predict the manifestation of adverse effects and the response to treatment in CRC patients. Herein, we provide a systematic review of epidemiological studies investigating epigenetic biomarkers in CRC patients receiving neoadjuvant or adjuvant therapy, and their potential role for the prediction of outcomes and response to treatment. With this aim in mind, we identified several epigenetic markers in CRC patients who received surgery with adjuvant or neoadjuvant therapy. However, none of them currently has the robustness to be translated into the clinical setting. Thus, more efforts and further large-size prospective studies and/or trials should be encouraged to develop epigenetic biomarker panels for personalized prevention and medicine in CRC cancer.

## 1. Introduction

In 2018, colorectal cancer (CRC) accounted for 823,000 and 1,026,000 new cancer cases in women and men, respectively, making it the third-most common cancer worldwide, and one of the main challenges for public health [1]. Although the introduction of colonoscopy has reduced CRC-related mortality in more-developed countries since the 1980s, the number of deaths has increased by approximately 60% between 1990 and 2013 [2]. Several genetic and environmental factors are crucial in the etiology of CRC by promoting the progressive accumulation of hallmark cancer traits in colon epithelial cells [3,4]. Beyond these factors, a lot of epigenetic modifications—including DNA methylation and aberrant expression of microRNAs (miRNAs)—commonly occur in early neoplastic lesions (e.g., aberrant crypt foci, adenomas and serrated polyps) and CRC, which in turn drive cancer initiation and formation along with gene mutations [5,6,7,8]. Epigenetic mechanisms can be defined as hereditary modifications that influence gene expression and the phenotype without changing the genotype. These molecular processes characterize the epigenome, which is dynamic in response to environmental signals, modifiable during normal cell differentiation and heritable in daughter cells [9]. In mammals, DNA methylation is regulated by the activity of three DNA methyltransferases (DNMTs) that catalyse the addition of a methyl group to the fifth carbon of the cytosine: while DNMT1 has a maintenance role, DNMT3a and 3b are de novo methylases. The DNA methylation process almost occurs within short CpG-rich regions (i.e., CpG islands), which typically contain almost 5–10 CpGs per 100 bp within the 5ʹ region of genes [10]. In general, cancer cells exhibit hypermethylation of CpG islands in gene promoters, which can result in transcriptional silencing of tumor suppressor genes, and hypomethylation of repetitive genetic elements, which can affect genomic stability [11]. Along with DNA methylation, miRNAs are endogenous, short, noncoding molecules that participate in post-transcriptional gene expression by suppressing the translation of protein-coding genes or by cleaving several messenger RNAs [12]. Indeed, miRNAs are important molecular mechanisms that can control physiological processes (i.e., cell growth, differentiation, stress response, and tissue remodeling) [13,14,15] and that play a key role in many disease states including CRC [16]. Particularly, in colon epithelium, both genomic and epigenomic instability can promote the accumulation of gene mutations and epigenetic modifications either in oncogenes or tumor suppressor genes, leading to the malignant transformation of colon cells [17,18,19]. Gene expression is also regulated by dynamic post-translational histone modifications, which may disrupt contacts within and between nucleosomes, recruit non-histone proteins, and affect the chromatin structure [20]. However, their effects depend on the type of modification, the type of amino acid involved and its position in the histone tail. Specifically, histone (de)acetylation and (de)methylation have been well-characterized in CRC patients and extensively reviewed elsewhere [21,22]. 

Although these molecular features are commonly shared between colon cancer and rectal cancer, the location of the tumor and its stage have implications for treatment. In general, surgery is the gold standard for the management of patients with non-metastasized CRC. In the last two decades, systemic approaches have been also developed, with strides in the application of the neoadjuvant therapy for rectal cancer and the adjuvant therapy for colon cancer [23]. In rectal cancer, neoadjuvant therapy—given as short-course radiotherapy followed by surgery or as chemoradiotherapy with 5‑fluorouracil (5-FU) or capecitabine—is recommended for intermediate-stage and advanced-stage patients [23]. While there are no accepted neoadjuvant therapies for colon cancer, the international guidelines recommend adjuvant chemotherapy for high-risk stage II (according to the Union for International Cancer Control (UICC) classification) and for all the stage III tumors [24]. Adjuvant treatment is generally given as a combination of 5‑FU plus oxaliplatin: as in the XELOX protocol for oral administration (capecitabine and oxaliplatin), or as in the FOLFOX4 protocol for intravenous administration (leucovorin, 5‑FU and oxaliplatin) [23]. In rectal cancer patients, adjuvant chemotherapy with fluoropyrimidines can be given only in those who did not receive preoperative treatment and in the presence of specific risk factors, such as positive resection margins, perforation in the tumor area or defects in the mesorectum [23]. In the metastatic disease, combinations of leucovorin, 5‑FU, and either oxaliplatin (FOLFOX protocol) or irinotecan (FOLFIRI protocol) represent the first-line therapies. More recently, the FOLFOXIRI protocol—the combination of leucovorin, 5‑FU, oxaliplatin and irinotecan—has been revealed to be efficacious [25]. 

Each of these treatments can be associated with certain adverse effects and different response rates. It has been recognized that patients receiving neoadjuvant chemoradiotherapy or radiotherapy often exhibit pelvic floor problems [26], with erectile dysfunction in men and dyspareunia in women [27,28]. With regard to chemotherapy, 5-FU is typically well tolerated whereas oxaliplatin and irinotecan are often associated with neutropenia and diarrhea [23]. Despite difficulties in creating an in vitro model, some studies have reported that neoadjuvant and adjuvant therapies might affect epigenetic mechanisms in colorectal cancer cells, determining the response to treatment [29,30]. In line with this evidence, several epidemiological studies have investigated epigenetic signatures—especially DNA methylation and miRNA expression—that might guide personalized treatments for subgroups of CRC patients in the forthcoming future [29,30]. However, until now, there have been no biomarkers to predict the manifestation of adverse effects and the response to treatment in these patients. Herein, we provide a systematic review of epidemiological studies investigating epigenetic biomarkers in CRC patients receiving neoadjuvant and/or adjuvant therapies, and their potential role for the prediction of outcomes and the response to treatment.

## 2. Literature Search

### 2.1. Search Strategy and Selection Criteria

Two of the Authors (AM and MB) performed a systematic literature search in the PubMed-Medline, EMBASE, and Web of Science databases from inception to January 2019, with no restriction on the language of publication. The following terms were used: (“Neoadjuvant Therapy” OR “Adjuvant Chemotherapy”) AND (“Colorectal Neoplasms” OR “Rectal Neoplasms” OR “Colonic Neoplasms”) AND (“DNA Methylation” OR “Methylation” OR “MicroRNAs” OR “Histone Code”). The Authors also searched the reference lists of selected articles to identify all relevant studies. The preferred reporting items for systematic reviews and meta-analysis (PRISMA) guidelines were followed [31] (Appendix A).

### 2.2. Selection Criteria and Data Collection 

Next, two of the Authors (AM and MB) independently selected the retrieved studies if they were consistent with the following criteria: (i) observational and experimental epidemiological studies (ii) on CRC, colon cancer or rectal cancer patients (iii) receiving surgery with neoadjuvant or adjuvant therapies, (iv) which focused on the relationship of DNA methylation, miRNA expression or histone modifications with overall survival, disease-specific survival, disease-free survival, recurrence-free survival, and response to treatment. In contrast, (i) systematic reviews; (ii) abstracts and unpublished studies; (iii) and those with no treated patients were excluded. From each study, the Authors (AM and MB) independently extracted the following information: first Author’s last name, year of publication, the total number of participants, type of cancer, type of treatment and the number of treated patients, sample source, epigenetic biomarker, and main findings. Any inconsistencies between the two Authors in the study selection and data extraction phases were resolved through discussion with a third Author (AA).

## 3. Results

### 3.1. Study Characteristics

Figure 1 shows the PRISMA flow diagram of study selection. After removing duplicates, a total of 55 articles were retrieved from the databases: five reviews were excluded after reading titles and/or abstracts, while 50 articles underwent full-text screening. According to selection criteria, we excluded one study with no epidemiological design, one study without treated group, and two studies that did not investigate epigenetic mechanisms. Six relevant studies that met the selection criteria were included from the reference lists of selected articles, and therefore the systematic review included a total of 52 studies published since 2003.

Overall, 20 studies were conducted on CRC patients, whereas 23 and 9 studies analyzed rectal cancer or colon cancer, respectively. Among them, 30 studies recruited CRC patients (*n* = 21) or colon cancer patients (*n* = 9) receiving adjuvant therapy, whereas 18 studies were conducted on rectal cancer patients receiving neoadjuvant therapy. Other studies recruited CRC (*n* = 1) or rectal cancer patients (*n* = 3) receiving either neoadjuvant therapy or adjuvant therapy. With respect to epigenetic biomarkers, 26 studies investigated DNA methylation changes in patients receiving neoadjuvant therapy (*n* = 6), adjuvant therapy (*n* = 18), or both (*n* = 2). Instead, other 26 studies investigated miRNA expression in patients receiving neoadjuvant therapy (*n* = 12), adjuvant therapy (*n* = 12), or both (*n* = 2). The majority of these studies analyzed epigenetic biomarkers in tumor tissues (*n* = 47), whereas only five studies used serum samples. No studies examined histone modifications and their association with response to adjuvant or neoadjuvant therapies.

### 3.2. DNA Methylation in Patients Receiving Adjuvant Therapy

Methylation markers in patients receiving adjuvant therapy—including gene-specific methylation, CpG island methylator phenotype (CIMP), and long interspersed nuclear element-1 (LINE-1) methylation—are discussed in the paragraphs below and the main findings of selected studies are displayed in Table 1, Table 2 and Table 3.

#### 3.2.1. Gene-Specific Methylation

To our knowledge, Nagasaka and colleagues were the first investigating the effect of DNA methylation in CRC patients receiving adjuvant therapy [32]. They found that patients with unmethylated O6-methylguanine-DNA methyltransferase (MGMT) promoter had a higher risk of recurrence within 36 months than those with MGMT hypermethylation. Interestingly, among patients with unmethylated MGMT promoter, those who received adjuvant chemotherapy had an increased risk of recurrence compared with untreated patients [32]. Although the mechanisms underpinning this relationship are currently unclear, the Authors concluded that MGMT hypermethylation might be associated with a lower risk of recurrence in CRC patients receiving adjuvant chemotherapy. A plausible explanation of this relationship is that the silencing of MGMT via promoter methylation, followed by decrease in its DNA repair activity, might enhance the effectiveness of chemotherapy [32]. However, molecular mechanisms that increased the risk of recurrence in patients with unmethylated MGMT promoter are yet to be clarified. In contrast to the above group, Chen and colleagues found no significant association between MGMT methylation and prognosis in CRC patients [33]. They also investigated the effect of methylation of the Adenomatous polyposis coli (APC) promoter—a tumor suppressor gene encoding a protein that regulates cell adhesion—demonstrating its association with lower risk of all-causes and CRC-related mortality [33]. Notably, germline APC mutations are one of the main cause of familial adenomatous polyposis, while somatic APC mutations are evident in eight out of 10 sporadic CRCs [34]. However, no interaction between 5-FU adjuvant chemotherapy and MGMT or APC methylation was reported [33].

Another gene that has been investigated is MutL homolog 1 (MLH1), a DNA mismatch repair gene. In 2008, Ide and colleagues first demonstrated that MLH1 expression was down-regulated in CRC tumors that exhibited MLH1 promoter methylation [35]. Next, among patients receiving 5-FU-based adjuvant therapy, they observed that those with reduced MLH1 expression levels had a significantly longer disease-free survival [35]. Methylation of MLH1 promoter can cause defect of the mismatch repair system along with germline mutations in mismatch repair genes including MLH1, MutS homolog 2 (MSH2), MSH6, and Postmeiotic Segregation Increased 2 (PMS2). With this in mind, Sinicrope and colleagues identified two subtypes of colon cancer deficient in mismatch repair based on mismatch repair status and detection of B-Raf Proto-Oncogene (BRAF) V600E or mutations in KRAS Proto-Oncogene (KRAS) [36]. However, the time of disease-free survival was similar between patients deficient in mismatch repair and those without BRAF V600E or KRAS mutations [36]. Over recent years, other methylation markers have been extensively debated. In 2014, Perez-Carbonell and colleagues evaluated the methylation status of Septin 9 (SEPT9), Aristaless-Like Homeobox 4 (ALX4), Twist-related protein 1 (TWIST1), Insulin Like Growth Factor Binding Protein 3 (IGFBP3), Growth Arrest Specific 7 (GAS7), and miR137 in CRC patients, and its association with clinicopathological characteristics [37]. While all genes were hypermethylated in tumor tissues compared with normal adjacent mucosa, only IGFBP3 methylation shortened disease free survival in CRC patients [37]. Interestingly, patients with IGFBP3 hypermethylation did not benefit from 5FU-based adjuvant therapy [37]. However, biochemical and molecular mechanisms that might explain this relationship are currently unknown. In the same year, Heitzer and colleagues aimed to identify novel predictive and prognostic biomarkers in patients randomized to adjuvant chemotherapy with 5-FU and leucovorin or surveillance only [38]. Their survival analysis pointed out that patients receiving adjuvant therapy with no methylation of a combined set of three genes (Protocadherin 10, PCDH10; Secreted Protein Acidic And Rich In Cysteine; SPARC; Ubiquitin C-Terminal Hydrolase L1; UCHL1) had longer overall survival than those with hypermethylation [38]. In the surveillance group, on the contrary, unmethylated genes were associated with shorter disease-free survival and overall survival [38]. Previous studies explained the involvement of these genes in tumor formation and progression. PCDH10 has been recently indicated as a tumor suppressor gene due to its role in cell cycle, tumor progression and metastasis [39,40]. Although still debated, SPARC appeared a crucial driver of tumor angiogenesis, cell proliferation and migration [38]. UCHL1 was indicated as a tumor suppressor gene [41], and its methylation might be a prognostic markers in several cancers [42]. In CRC patients receiving adjuvant chemotherapy, Chang and colleagues demonstrated that Homeobox Protein NK-6 Homolog A (NKX6.1) methylation levels were significantly higher in tumor tissues than in adjacent normal mucosa [43]. Next, they showed that NKX6.1 hypermethylation reduced overall survival and disease-free survival [43]. Accordingly, NKX6.1 hypermethylation has been previously reported in cervical cancer [44], acute lymphoblastic leukemia [45], and gastric cancer [46]. In colon cancer patients, Pfütze and colleagues evaluated methylation status of (hyaluronoglucosaminidase 2 protein) HYAL2, a gene encoding HYAL2 with an altered expression in several cancers [47]. In patients receiving 5-FU based adjuvant therapy, HYAL2 hypomethylation improved overall survival and progression-free survival. In patients not receiving adjuvant chemotherapy, on the contrary, HYAL2 hypomethylation seemed to be associated with worse overall survival [47].

#### 3.2.2. CpG Island Phenotype

In 2003, Van Rijnsoever and colleagues investigated the potential of CIMP for predicting prognosis in CRC patients treated with 5-FU adjuvant chemotherapy [48]. In general, CIMP is characterized DNA hypermethylation of genes involved in controlling cell growth and survival. Positive or high-level CIMP (CIMP+ or CIMP-high, according to different classifications) were associated with specific clinicopathological features of CRC, such as mucinous histology of the proximal colon and BRAF mutation [29]. In the study by Van Rijnsoever and colleagues, CIMP+ (based on a panel of three methylation markers including Cyclin Dependent Kinase Inhibitor 2A, Methylated-IN-Tumour locus 2, and Multi Drug Reactivity 1 genes; CDKN2A, MINT-2 and MDR1, respectively) was associated with worse prognosis in CRC patients treated with surgery alone [48]. By contrast, CIMP+ patients receiving adjuvant therapy exhibited improved survival, while no long-term benefits were evident in CIMP- patients [48]. Min and colleagues corroborated this evidence using the Methylight assay to evaluate a panel of five methylation markers (i.e., Calcium Voltage-Gated Channel Subunit Alpha1 G, Insulin Like Growth Factor 2, Neurogenin 1, Runt-Related Transcription Factor 3, and Suppressor Of Cytokine Signaling 1 genes; CACNA1G, IGF2, NEUROG1, RUNX3, and SOCS1, respectively) in CRC patients [49]. They first demonstrated that CIMP-high was also associated with high frequencies of MGMT methylation and microsatellite instability [49]. Moreover, CIMP+ patients receiving adjuvant therapy exhibited increased recurrence-free survival than patients treated with surgery alone, while no benefits from adjuvant therapy were evident in CIMP- or CIMP-low patients [49]. In colon cancer, Shiovitz and colleagues—using the same panel of methylation markers—corroborated that patients with CIMP+ had shorter overall survival than those with CIMP- tumors [50]. They also demonstrated that CIMP+ patients treated with the FOLFIRI protocol had better overall survival than those treated with 5-FU and leucovorin alone. However, benefits of this combination were not reported in patients with CIMP- tumors [50]. Conversely, Cohen and colleagues showed that CIMP+ status based on the same panel of methylation markers did not affect overall survival in CRC patients, but it was associated with right-side tumor location and BRAF mutations [51]. Jover and colleagues used pyrosequencing to analyze an alternative panel of five markers (i.e., CACNAG1, SOCS1, RUNX3, NEUROG1, and MLH1) in CRC patients receiving 5-FU adjuvant therapy [52]. They found an interaction between 5-FU treatment and CIMP status, demonstrating that adjuvant therapy improved disease-free survival in CIMP- but not in CIMP+ patients [52]. Interestingly, in patients not receiving adjuvant therapy, CIMP status was the only predictor of disease-free survival [52]. In 2013, Han and colleagues analyzed CIMP status based on an extended panel of eight markers (i.e., CACNA1G, CDKN2A, IGF2, MLH1, NEUROG1, RUNX3, SOCS1, and Cellular Retinoic Acid Binding Protein 1 – CRABP1) in CRC patients who received adjuvant therapy with the FOLFOX protocol [53]. They showed that CIMP status was associated with microsatellite instability but not with disease-free survival. However, methylation of NEUROG1 and CDKN2A increased the risk of recurrence [53]. Bae and colleagues revised the eight-marker panel for a new CIMP classification system that classified tumors into three different CIMP subtypes (i.e., CIMP-: 0–4 methylated markers, CIMP+1: 5–6 methylated markers and CIMP+2: 7–8 methylated markers) [54]. Among nearly 1400 CRC patients treated with surgery alone, those with CIMP+1 tumors reported lower cancer-specific survival and relapse-free survival than those with CIMP- and CIMP+2 tumors [54]. Next, the Authors confirmed this finding in a cohort of CRC patients treated with FOLFOX or XELOX protocols [54].

#### 3.2.3. Long Interspersed Nuclear Element-1 Methylation

LINE-1 sequences—a class of retrotransposons capable of independent and autonomous retro-transposition via RNA intermediate [55]—account for ≈ 18% of the human genome, with more than 500,000 copies. Given their abundance throughout the genome, methylation status of these sequences has been widely used as a surrogate marker of global methylation in aging and age-related disease [56,57,58,59,60,61,62,63]. Particularly, a meta-analysis showed that LINE-1 methylation levels were significantly lower in tissue samples from CRC patients than in healthy controls [64]. Indeed, hypomethylation of these elements might be affected by dietary habits and environmental exposures [65,66,67,68], thereby increasing the risk of chromosomal instability and aberrant genome function [69,70].

In 2011, Kawakami and colleagues investigated LINE-1 methylation level as a predictive and prognostic marker in CRC patients receiving surgery alone or in combination with 5-FU adjuvant chemotherapy [71]. Since LINE-1 methylation correlated with CIMP status [72], the Authors recruited a subgroup of CIMP- patients. In patients treated with surgery alone, LINE-1 hypomethylation was associated with worse prognosis, whereas it conferred a survival benefit in patients receiving adjuvant therapy [71]. Benefits from adjuvant chemotherapy were not evident in patients with LINE-1 hypermethylation [71]. Next, Chen and colleagues confirmed that LINE-1 methylation was associated with clinicopathological features and recurrence-free survival in CRC patients receiving adjuvant therapy based on the FOLFOX protocol [73]. Similarly, Lou and colleagues reported lower LINE-1 methylation level in colon cancer patients receiving adjuvant therapy with post-therapeutic recurrence than in those without recurrence [74]. Their multivariable analysis confirmed LINE-1 hypomethylation as an independent risk factor of post-therapeutic recurrence. In addition, patients with LINE-1 hypomethylation had reduced disease free survival in the whole cohort and in those with post-therapeutic recurrence after 6 months [74].

### 3.3. DNA Methylation in Patients Receiving Neoadjuvant Therapy

Methylation markers in patients receiving neoadjuvant therapy—including gene-specific methylation, CpG island methylator phenotype, and global or genome-wide methylation—are discussed in the paragraphs below and the main findings of selected studies are summarized in Table 4, Table 5 and Table 6.

#### 3.3.1. Gene-Specific Methylation

In 2010, De Maat and colleagues were the first investigating the prognostic value of DNA methylation in rectal cancer patients receiving neoadjuvant chemoradiation [75]. They found that methylation status of several MINT loci (i.e., MINT3 hypermethylation and MINT17 hypomethylation) reduced the risk of recurrence [75]. Since aberrant methylation of these loci has been also observed in CRC, several groups included them in some alternative CIMP panels. In the following years, few studies evaluated the role of DNA methylation in predicting response to neoadjuvant chemoradiation. Ebert and colleagues demonstrated the association of hypermethylation of the Transcription Factor Activating Protein 2 Epsilon (TFAP2E) gene with non-response to neoadjuvant chemoradiation in rectal cancer patients [76]. In contrast, the odds of adequate response were 6-fold higher in patients with TFAP2E hypomethylation compared with the whole cohort. Next, Molinari and colleagues profiled 24 tumor suppressors genes in tissues from rectal cancer patients receiving neoadjuvant chemoradiotherapy [77]. Compared with adjacent normal tissues, tumor samples exhibited hypermethylation of Estrogen Receptor 1, Cadherin 13, Retinoic Acid Receptor Beta, Cell Surface Adhesion Molecule, and Adenomatous Polyposis Coli (ESR1, CDH13, RARB, IGSF4, and APC) genes [77]. With respect to response to neoadjuvant therapy, however, only Tissue Inhibitor of Metalloproteinases 3 (TIMP3) methylation status was significantly different across four tumor regression grade classes. Particularly, non-responders to neoadjuvant therapy displayed TIMP3 hypomethylation than responders [77]. While the above research groups analyzed DNA methylation in tumor tissues, Sun and colleagues investigated MGMT methylation in cell-free DNA from serum of rectal cancer patients, and its relationship with response to 5-FU-based neoadjuvant chemoradiation [78]. Interestingly, they found that MGMT hypermethylation was associated with improved response to treatment and higher regression compared with MGMT hypomethylation [78].

#### 3.3.2. CpG Island Methylator Phenotype

In 2012, Jo and colleagues used methylation specific PCR to examine a panel of 5 CIMP markers (i.e., CACNA1G, IGF2, NEUROG1, RUNX3, and SOCS1) in rectal cancer patients receiving 5-FU-based neoadjuvant chemoradiation [79]. Contrary to other research groups, they reported no association with clinicopathological features, and KRAS or BRAF mutations. Although CIMP+ patients exhibited worse 3-year and 5-year disease-free survival, no association with response to neoadjuvant chemoradiation was evident [79]. Using the same CIMP panel on rectal cancer patients receiving adjuvant or neoadjuvant therapy, Kohonen-Corish and colleagues showed no association of CIMP+ status with overall survival [80]. However, they observed that combination of CDKN2A methylation and KRAS mutations independently predicted the risk of recurrence, thereby reducing overall and cancer-specific survival [80].

#### 3.3.3. Global and Genome-Wide Methylation

To our knowledge, few studies examined global or genome-wide methylation in rectal cancer patients receiving neoadjuvant chemoradiation. In 2014, Tsang and colleagues compared global DNA methylation between pre- and post-treatment resection specimens, using immunohistochemical staining followed by image analysis [81]. They observed a significant reduction of global DNA methylation in the majority of patients. Moreover, global methylation of pre-treatment specimens was significantly correlated with tumor stage and regression [81]. In the same year, Gaedcke and colleagues used a genome methylation CpG island array to identify differentially methylated regions (DMRs) that discriminate patients according to their response to treatment [82]. They first identified 20 highly discriminative DMRs in a discovery cohort. Next, in two additional validation cohorts, they confirmed and validated 10 DMRs that allowed the discrimination of patients with different prognosis [82].

### 3.4. MicroRNA in Patients Receiving Adjuvant Therapy

In the last decade, several studies investigated miRNA signatures in CRC patients receiving adjuvant therapy, and their main findings are displayed in Table 7. In 2008, Schetter and colleagues were the first to examine miRNA expression in colon cancer patients receiving 5-FU based adjuvant chemotherapy [83]. They first compared miRNA profiles between tumor tissues and adjacent normal mucosa in a test cohort of Caucasians, identifying 37 differentially expressed miRNAs [83]. Next, they confirmed five miRNAs (i.e., miR-20a, miR-21, miR-106a, miR-181b, and miR-203) that were enriched in tumor tissues from a validation cohort of Asians. Among these miRNAs, miR-21 upregulation was associated with poor survival and poor therapeutic response in both cohorts [83]. This finding was corroborated by Oue and colleagues, which examined miR-21 expression in colon cancers from two independent cohorts of Caucasians and Asians [84]. In both cohorts, upregulation of miR-21 was associated with poor survival independent of clinicopathological features [84]. Interestingly, patients with miR-21 upregulation did not benefit from the adjuvant chemotherapy. In the Asian cohort, patients receiving 5-FU based adjuvant chemotherapy with miR-21 downregulation showed better survival than those with upregulation [84]. More recently, Coebergh van den Braak and colleagues evaluated the prognostic value of a 2-miRNA signature (i.e., let-7i and miR-10b) in colon cancer patients treated with (stage III patients) or without (stage I-II patients) adjuvant chemotherapy [85]. Notably, 2-miRNA signature predicted hepatic recurrence in stage I-II patients. In the same group, the combination with miR-30b also predicted distant metastasis. In contrast, this 2-miRNA signature had no prognostic value in stage III patients receiving adjuvant therapy [85]. In 2011, Ma and colleagues applied a miRNA microarray to CRC patients receiving adjuvant chemotherapy, demonstrating that miR-150 was downregulated in tumors compared with adjacent normal tissues [86]. In addition, they observed that low miR-150 expression levels were associated with reduced overall survival and worse response to treatment [86]. A similar approach was applied by Perez-Carbonell and colleagues that profiled miRNAs in CRC patients receiving 5-FU based adjuvant chemotherapy, using the Affymetrix miRNA expression array [87]. They found that miR-320e was upregulated in patients with recurrence compared with those without recurrence. Moreover, miR-320e upregulation was associated with poor overall and disease-free survival and overall survival [87]. Similarly, Zhang and colleagues used miRNA microarrays to compare CRC tissues and normal adjacent mucosa, identifying 35 differentially expressed miRNAs [88]. Next, they developed a 6-miRNA-based classifier and evaluated its prognostic and predictive values in three independent sets of patients [88]. Based on this classifier, patients were classified according to their risk of disease progression. Five-year disease-free survival was higher in low risk patients compare with those at high risk in either set of patients analyzed. Interestingly, patients with high risk of disease progression exhibited a better response to 5-FU based adjuvant chemotherapy [88]. To and colleagues examined miR-519c expression in CRC patients receiving adjuvant chemotherapy, and its association with ATP Binding Cassette Subfamily G Member 2 (ABCG2) expression [89]. Notably, miR-519c levels were positively correlated with ABCG2 expression, and the most of CRC samples from non-responders exhibited miR-519c downregulation [89]. Diaz and colleagues compared expression levels of several miRNAs (i.e., including miR-200a, miR-200b, miR-200c, miR-141, and miR-429) between tumor and adjacent normal tissues in CRC patients receiving 5-FU or XELOX treatment [90]. They demonstrated that upregulation of miR-200a, miR-200c and miR-429 was associated with better survival. Among patients receiving 5-FU based adjuvant therapy, upregulation of miR-200a, miR-200c, miR-141, or miR-429 was positively associated with overall and disease-free survival [90]. Li and colleagues compared miR-215 expression between CRC patients with or without recurrence within 3 years after surgery [91]. Patients with recurrence exhibited lower miR-215 expression level than those without recurrence. Moreover, the Authors observed that miR-215 downregulation was associated with worse response to 5-FU adjuvant therapy [91]. Dou and colleagues specifically aimed to evaluate genes and miRNAs that allowed them to identify CRC patients who could benefit from adjuvant chemotherapy [92]. They revealed 17 differentially expressed miRNAs between responders and non-responders. These results, along with those from gene expression analysis, allowed to identify 3 genes (i.e., Aquaporin-9, Special AT-Rich Sequence-Binding Protein 2, Wnt Inhibitory Factor 1; AQP9, SATB2, and WIF1, respectively) that were downregulated by the differentially expressed miRNAs [92]. Differently from above studies, Conev and colleagues evaluated whether expression of four miRNAs (i.e., miR-17, miR-21, miR-29a, and miR-92) in serum samples might predict the risk of recurrence in colon cancer patients receiving adjuvant chemotherapy [93]. They demonstrated that miR-17, miR-21 and miR-92 were upregulated in patients with recurrence, with AUC of 0.844, 0.948, and 0.935, respectively. Notably, a test based on the four miRNAs allowed to discriminate patients at higher risk of recurrence, with a sensitivity of 83.3% and a specificity of 85.7% [93]. In line with this evidence, Liu and colleagues examined the potential of serum exosomal miRNAs to predict the risk of recurrence and the response to adjuvant chemotherapy in colon cancer patients [94]. Using RNA sequencing, they identified 145 differentially expressed miRNAs between patients with or without recurrence. Particularly, downregulation of miR-4772-3p was associated with an increased risk of recurrence, with an AUC of 0.72. This evidence was confirmed in patients receiving adjuvant therapy [94].

### 3.5. MicroRNA in Patients Receiving Neoadjuvant Therapy

With respect to rectal cancer patients receiving neoadjuvant chemoradiation, other research groups investigated the predictive and prognostic values of several miRNAs, and their main findings are summarized in Table 8. In 2012, Gaedcke and colleagues applied a miRNA microarray to compare miRNA expression between tumor tissues and adjacent normal mucosa in rectal cancer patients receiving neoadjuvant chemoradiation [95]. A preliminary analysis identified a set of miRNA (i.e., miR-492, miR-542-5p, miR-584, miR-483-5p, miR-144, miR-2110, miR-652, miR-375, miR-147b, miR-148a, miR-190, miR-26a/b, and miR-338-3p) that were differently expressed between rectal and colon cancer tissues [95]. In an independent cohort, the Authors demonstrated that miR-135b expression correlated with tumor regression grade, disease-free survival and cancer-specific survival [95]. In the same years, Drebber and colleagues examined expression of miR-21, -143 and -145 in rectal cancer receiving neoadjuvant chemoradiotherapy: among these miRNAs, miR-21 was upregulated in tumor tissue than in adjacent normal mucosa [96]. With respect to changes induced by neoadjuvant chemoradiation, miR-21 was downregulated while miR-143 and miR-145 were upregulated in post-treatment tissues compared with pre-treatment tissues. In addition, post-treatment miR-145 downregulation seemed to be associated with worse response to neoadjuvant therapy [96]. In 2012, independent research groups begun to investigate miRNA-signatures that allowed to predict response to neoadjuvant chemoradiation by comparing responders with non-responders. Svoboda and colleagues applied a miRNA microarray to rectal cancer patients receiving neoadjuvant chemoradiation classified as responders or non-responders [97]. They identified eight differentially expressed miRNAs, as such miR-215, miR-190b and miR-29b-2 were upregulated in non-responders, while let-7e, miR-196b, miR-450a, miR-450b-5p and miR-99a were upregulated in responders. Interestingly, this 8-miRNA signature allowed to correctly classify 90% of non-responders [97]. Kheirelseid and colleagues performed a microarray analysis on pre-treatment tissues of rectal cancer patients receiving neoadjuvant chemoradiation [98]. They identified a 3-miRNA signature (i.e., miR-16, miR-590-5p and miR-153) that predicted incomplete response to neoadjuvant therapy [98]. They also proposed a 2-miRNA signature (i.e., miR-519c-3p and miR-561) that predicted poor response to neoadjuvant therapy with an accuracy of 100% [98]. Della Vittoria Scarpati and colleagues used the same approach on rectal cancer patients receiving neoadjuvant therapy classified by their tumor regression grade [99]. Accordingly, they first identified 14 differentially expressed miRNAs in patients with complete response [99]. Next, they validated 11 upregulated (miR-1183, miR-483-5p, miR-622, miR-125a-3p, miR-1224-5p, miR-188-5p, miR-1471, miR-671-5p, miR-1909, miR-630, miR-765) and two downregulated (miR-1274b, miR-720) miRNAs in the same patients. Notably, miR-622 and miR-630 showed 100% sensitivity and specificity in discriminating patients with complete response [99]. In line with this finding, Campayo and colleagues identified eight miRNAs (i.e., let-7b, let-7e, miR-21, miR-99b, miR-183, miR-328, miR-375 and miR-483-5p) that were differentially expressed across different tumor regression grades [100]. In a validation set, they observed that miR-21, miR-99b and miR-375 were associated with disease-free and overall survival [100]. Accordingly, the Authors concluded that the combination of miR-21, miR-99b and miR-375 allowed to discriminate patients with complete response to neoadjuvant therapy [100]. In 2014, Lopes-Ramos and colleagues profiled miRNAs using RNA-sequencing in pre-treatment rectal cancer tissues from patients receiving neoadjuvant chemoradiation [101]. They first identified four differentially expressed miRNAs between complete and incomplete responders in a training set [101]. Next, they confirmed that miR-21-5p was upregulated in complete responders of a validation set. Interestingly, miR-21-5p expression level allowed the prediction of complete response to neoadjuvant therapy with a sensitivity of 78% and a specificity of 86% [101]. In 2016, Caramés and colleagues measured miR-31 expression level in rectal cancer tissues from patients receiving neoadjuvant chemoradiation, using real-time PCR [102]. They observed higher miR-31 level in patients with poor response than in those with complete response. Accordingly, miR-31 upregulation was associated with poor pathological response and worse overall survival [102]. Similarly, D’Angelo and colleagues demonstrated that miR-194 was overexpressed in patients who responded to neoadjuvant chemoradiation [103]. More recently, Du and colleagues identified 41 differentially expressed miRNAs between complete and incomplete responders to neoadjuvant therapy [104]. Among these miRNAs, miR-548c-5p/miR-548d-5p and miR-663a regulated genes associate with rectal cancer, thereby modulating the complete response to neoadjuvant chemoradiation [104]. Luo and colleagues measured the expression level of miR-519b-3p in patients receiving neoadjuvant therapy, demonstrating a positive correlation with response to treatment [105]. Interestingly, functional analysis suggested that miR-519b-3p was directly involved in response to neoadjuvant chemoradiation in an ARID4B-dependent way [105]. In 2016, Yu and colleagues used a microarray to examine miRNA profiles in pre-treatment tissue and serum samples from rectal cancer patients receiving neoadjuvant chemoradiation [106]. In a validation set, miR-345 upregulation was associated with a worse response to treatment either in tissue or serum. Accordingly, miR-345 downregulation in serum was associated with better recurrence-free survival [106]. In line with this evidence and results from above studies, Menéndez and colleagues evaluated serum expression level of miR-21—one of the most investigated miRNAs in this field of research—in rectal cancer patients [107]. Interestingly, serum miR-21 downregulation was associated with high risk of recurrence and death. Indeed, a Cox regression analysis demonstrated that miR-21 expression was an independent predictor of overall survival [107]. Recently, Hiyoshi and colleagues analyzed 18 serum miRNAs in rectal patients receiving neoadjuvant chemoradiation, using real-time PCR [108]. Among these miRNAs, miR-143 was associated with response to treatment, with higher expression levels in responders than in non-responders. A multivariate analysis confirmed serum miR-143 expression as an independent predictor of pathological response [108].

## 4. Discussion

Over the past decades, CRC has become one of the most common cancers with an associated mortality that remains high in spite of substantial advances in treatment and patient management [1]. Moreover, each treatment is associated with specific adverse effects and complications [23]. For instance, treatment with oxaliplatin or irinotecan is often associated with adverse effects, such as neutropenia and diarrhea, while chemotherapy with 5‑fluorouracil is usually well tolerated [23]. Moreover, strategies aiming to improve the response to neoadjuvant therapy by intensifying the chemoradiotherapy regimen (e.g., the combination of 5-FU and oxaliplatin with radiotherapy) did not exhibit clear survival benefit, and even increased toxicity with persisting chronic complications in many patients [23]. Thus, despite strides in the application of adjuvant and/or neoadjuvant therapies, there is currently an unresolved need for distinguishing patients who might respond to treatment from those who do not. Although several molecular biomarkers have been proposed for predicting the response to treatment, more efforts should be made to tailor therapies for patients with specific molecular features [23]. 

In this *scenario*, our study provides the first systematic review of epidemiological studies investigating the predictive and prognostic value of epigenetic biomarkers in CRC patients receiving neoadjuvant and/or adjuvant therapies. We summarized that gene-specific methylation has been associated with risk of recurrence, response to treatment, and survival in patients receiving either adjuvant or neoadjuvant therapy. For instance, demethylation of MGMT promoter in tumor tissue seemed to be associated with higher risk of recurrence in general, and specifically in patients who received adjuvant chemotherapy. By contrast, those with MGMT hypermethylation had a reduced risk of recurrence [32]. A plausible explanation is that the repression of MGMT expression by promoter methylation might cause a decrease in its DNA repair activity, hereby enhancing the effectiveness of chemotherapy [32]. In line with this evidence, MGMT hypermethylation of cell-free DNA from serum was associated with improved response to treatment and higher regression in rectal cancer patients who received 5-FU-based neoadjuvant chemoradiation [78]. However, these findings need to be confirmed by further research, since other groups found no significant association between MGMT methylation and prognosis in CRC patients [33]. Another gene that has been associated with CRC prognosis and response to adjuvant therapy was MLH1. Particularly, patients receiving 5-FU-based adjuvant therapy with MLH1 hypermethylation and reduced expression levels had a significantly longer disease-free survival than their counterpart [35]. Indeed, methylation of MLH1 promoter can cause a defect in the mismatch repair system along with germline mutations in mismatch repair genes [36]. Based on these findings and also those reported by other studies, several panels of methylation markers (i.e., CIMP) have been proposed for predicting prognosis in CRC patients. In general, CIMP+ status seemed to be associated with worse disease-free and overall survival in either CRC or colon cancer patients; however, more benefits from the adjuvant therapy were reported in CIMP+ than in CIMP- patients. In contrast, studies investigating CIMP panels in rectal cancer patients receiving neoadjuvant therapy demonstrated no association with response to treatment. Toyota and colleagues were the first reporting the classification of CIMP status about twenty years ago [109], but there is as yet no consensus on the definitive panel for classifying it. Indeed, the comparison of several CIMP panels found significant differences in CIMP positivity determined by different panels [110]. This variability should be taken into account when comparing findings from different studies and before proposing a certain panel for risk stratification. 

With respect to miRNAs, independent studies conducted on Caucasian and Asian populations demonstrated the relationship between miR-21 upregulation in tumor tissue and poor survival of CRC patients receiving adjuvant therapy [83,84]. Instead, patients with miR-21 upregulation did not benefit from the adjuvant chemotherapy [84]. In rectal cancer patients who received neoadjuvant chemoradiotherapy, the analysis of three different miRNAs revealed that miR-21 was upregulated in tumor tissue but not in the adjacent normal mucosa [96]. However, patients with higher miR-21 levels exhibited better disease-free and overall survival than those with lower miR-21 expression [100]. In addition, the combination of miR-21 with miR-99b and miR-375 (i.e., the sum of their expression levels) allowed to discriminate patients with complete response to neoadjuvant therapy [100]. Consistently, RNA-sequencing of rectal cancer tissues from patients receiving neoadjuvant chemoradiation demonstrated that miR-21-5p—one of the mature sequences of miR-21—was upregulated in complete responders [101]. Indeed, the evaluation of its expression level allowed the prediction of complete response to neoadjuvant therapy with a sensitivity of 78% and a specificity of 86% [101]. In line with this evidence, expression level of miR-21 was further investigated in serum samples from rectal cancer patients [107]. This study demonstrated that downregulation of miR-21 was associated with higher risk of recurrence and death, as confirmed by a Cox regression analysis [107]. Beyond miR-21, other miRNA-signatures in the serum have been proposed to predict overall survival, risk of recurrence and response to treatment with high sensitivity and specificity. However, further large-size prospective studies are required to confirm the reliability and the robustness of these findings. 

In general, studies included in our systematic review showed high heterogeneity in terms of disease classification, methods for determining methylation, and tumor response evaluation. Compared with other tumors, several molecular features (e.g., DNA mutation, epigenetic signatures, and oncogenic pathway action) make CRC a heterogeneous disease with high intra-tumor heterogeneity [111]. In addition, we found significant heterogeneity in treatment schedules, which might lead to different response to treatment. The high heterogeneity between and within tumors limits the sensitivity and specificity of single biomarkers to predict response to treatment. Thus, in our opinion, considerable efforts are required to elucidate the efficacy of these markers for classifying CRC patients according to their response to treatment, and to combine these epigenetic signatures into wider panels of molecular and clinical markers. Moreover, our systematic review pointed out the lack of evidence on the effects of histone modifications on the response to adjuvant or neoadjuvant therapies. Since histone modifications have been associated with aberrant gene expression in cancer development and progression, their implication as potential biomarkers for response to therapies should be investigated in the future. 

In conclusion, our systematic review identified several epigenetic markers in CRC patients who received surgery with adjuvant or neoadjuvant therapy. However, none of them currently has the robustness to be translated into the clinical setting. While most studies generally focused on only one type of biomarker, an integrated approach of gene expression profiles, DNA methylation, and miRNAs might allow us to develop more accurate biomarker panels that can predict response to adjuvant or neoadjuvant therapy. With this in mind, more efforts and further large-size prospective studies and/or trials should be encouraged to develop epigenetic biomarker panels for personalized prevention and medicine in CRC cancer.

## Figures and Tables

**Figure 1 ijms-20-03842-f001:**
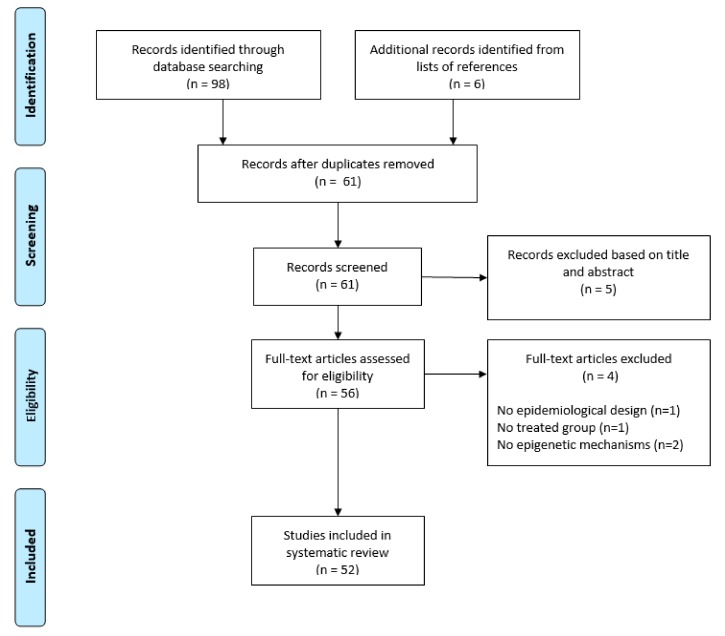
PRISMA 2009 Flow Diagram of study selection.

**Table 1 ijms-20-03842-t001:** Gene-specific methylation markers in patients receiving adjuvant chemotherapy.

First Author and Year	Tumor	Overall Population (Treated Patients)	Treatment	Samples	Markers	Methods	Response to Treatment	Other Findings
Nagasaka, T., 2003 [32]	CRC	90 (50)	5-FU	Tumor and normal tissues	MGMT	MS-PCR	In the unmethylated group, patients receiving adjuvant chemotherapy had greater risk of recurrence than those who did not receive chemotherapy	Patients with unmethylated MGMT were more likely to experience recurrence within 36 months than those with MGMT methylation
Chen, S.P., 2009 [33]	CRC	117 (61)	5-FU	Tumor tissues	APC and MGMT	MS-PCR	No interaction between adjuvant chemotherapy and methylation markers was evident	Patients with APC methylation had lower risk of death or CRC-related deaths. No significant association MGMT methylation and prognosis was evident
Ide, T., 2008 [35]	CRC	94 (35)	5-FU	Tumor and normal tissues	MLH1	MS-PCR	Among patients receiving adjuvant chemotherapy, those with low hMLH1 mRNA expression levels had longer disease-free survival	MLH1 mRNA expression levels were significantly lower in CRC tissues with MLH1 methylation compared with unmethylated hMLH1 tissues
Sinicrope, F.A., 2015 [36]	Colon cancer	2720 (2720)	mFOLFOX6 alone or with cetuximab	Tumor tissues	MLH1	MS-PCR	Not examined	Two subtypes of colon cancer deficient in mismatch repair were identified, based on mismatch repair status and detection of BRAFV600E or mutations in KRAS. However, time of disease-free survival was similar between patients deficient in mismatch repair and those without BRAFV600E or KRAS mutations
Perez-Carbonell, L., 2014 [37]	CRC	425 (425)	5-FU	Tumor tissues	SEPT9, TWIST1, IGFBP3, GAS7, ALX4 and miR137	Pyrosequencing	Patients with IGFBP3 hypermethylation did not benefit from adjuvant chemotherapy	Methylation levels of all genes analyzed were significantly higher in tumor tissues than in normal mucosa. IGFBP3 hypomethylation was an independent risk factor for poor disease-free survival
Heitzer, E., 2014 [38]	CRC	143 (71)	5-FU and leucovorin	Tumor tissues	PCDH10, SPARC, and UCHL1,	Real-time PCR	Patients receiving adjuvant therapy with no methylation of a combined set of three genes had longer overall survival than those with hypermethylation. In the surveillance group, unmethylated genes were associated with shorter disease-free survival and overall survival.	NA
Chang, S.Y., 2018 [43]	CRC	151 (151)	NA	Tumor and normal tissues	NKX6.1	MS-PCR	Patients receiving adjuvant chemotherapy with NKX6.1 methylation had shorter 5-year overall survival and disease-free survival than patients with no NKX6.1 methylation	NKX6.1 methylation was significantly higher in tumor tissues than in adjacent normal tissues
Pfütze, K., 2015 [47]	Colon cancer	232 (98)	5-FU	Tumor tissues	HYAL2	MALDI-TOF mass spectrometry	In patients receiving adjuvant chemotherapy, low methylation levels were associated with better overall survival and progression-free survival. In patients no receiving adjuvant chemotherapy, on the contrary, HYAL2 hypomethylation seemed to be associated with worse overall survival	NA

Abbreviations: CRC, Colorectal Cancer; 5-FU, 5 fluorouracil; MGMT, O6-methylguanine-DNA methyltransferase; MS-PCR, methylation-specific PCR; APC, Adenomatous polyposis coli; MLH1, MutL homolog 1; BRAF, B-Raf Proto-Oncogene; KRAS, KRAS Proto-Oncogene; SEPT9, Septin 9; ALX4, Aristaless-Like Homeobox 4; TWIST1, Twist-related protein 1; IGFBP3, Insulin Like Growth Factor Binding Protein 3; GAS7, Growth Arrest Specific 7; PCDH10, Protocadherin 10; SPARC, Secreted Protein Acidic And Rich In Cysteine; UCHL1, Ubiquitin C-Terminal Hydrolase L1; NA, Not-available; NKX6.1, Homeobox Protein NK-6 Homolog A; HYAL2, Hyaluronoglucosaminidase 2 protein; MALDI-TOF, Matrix Assisted Laser Desorption Ionization Time-of-Flight.

**Table 2 ijms-20-03842-t002:** CpG island methylator phenotype markers in patients receiving adjuvant chemotherapy.

First Author and Year	Tumor	Overall Population (Treated Patients)	Treatment	Samples	Markers	Methods	Response to Treatment	Other Findings
Van Rijnsoever, M., 2003 [48]	CRC	206 (103)	5-FU	Tumor tissues	CIMP (CDKN2A, MINT-2 and MDR1)	MS-PCR	CIMP+ patients receiving adjuvant therapy exhibited improved survival, while no long-term benefits were evident in CIMP- patients	CIMP+ was associated with worse prognosis in patients treated with surgery alone
Min, B.H., 2011 [49]	CRC	245 (124)	5-FU or capecitabine	Tumor tissues	CIMP (CACNA1G, IGF2, NEUROG1, RUNX3, and SOCS1)	MethyLight assay	CIMP+ patients receiving adjuvant therapy exhibited increased recurrence-free survival than patients treated with surgery alone, while no benefits from adjuvant therapy were evident in CIMP- or CIMP-low patients	CIMP-high was also associated with high frequencies of MGMT methylation and microsatellite instability
Shiovitz, S., 2014 [50]	Colon cancer	615 (316/299)	5-FU and leucovorin alone or with irinotecan	Tumor tissues	CIMP (CACNA1G, IGF2, NEUROG1, RUNX3, and SOCS1)	MethyLight assay	CIMP+ patients treated with the FOLFIRI protocol had better overall survival than those treated with 5-FU and leucovorin alone. However, benefits of this combination were not reported in patients with CIMP- tumors	CIMP+ had shorter overall survival than those with CIMP- tumors
Cohen, S.A., 2016 [51]	CRC	292 (144/148)	mFOLFOX6 or XELOX	Tumor tissues	CIMP (CACNA1G, IGF2, NEUROG1, RUNX3, and SOCS1)	MethyLight Assay	Not examined	CIMP+ status did not affect overall survival, but it was associated with right-side tumor location and BRAF mutations
Jover, R., 2011 [52]	CRC	302 (93)	5-FU	Tumor tissues	CIMP (CACNA1G, SOCS1, RUNX3, NEUROG1, and MLH1)	Pyrosequencing	Adjuvant therapy improved disease-free survival in CIMP- but not in CIMP+ patients. In patients not receiving adjuvant therapy, CIMP status was the only predictor of disease-free survival	NA
Han, S.W., 2013 [53]	CRC	322 (322)	FOLFOX	Tumor and normal tissues	CIMP (CACNA1G, CDKN2A, CRABP1, IGF2, MLH1, NEUROG1, RUNX3 and SOCS1)	MethyLight assay	Not examined	CIMP status was associated with microsatellite instability but not with disease-free survival. Methylation of NEUROG1 and CDKN2A increased the risk of recurrence
Bae, J.M., 2017 [54]	CRC	1370 (531/365/49)	5-FU and leucovorin, FOLFOX, or FOLFIRI	Tumor tissues	CIMP (CACNA1G, CDKN2A, CRABP1, IGF2, MLH1, NEUROG1, RUNX3 and SOCS1)	MethyLight assay	Among patients treated FOLFOX or XELOX protocols, CIMP+1 tumors reported lower cancer-specific survival and relapse-free survival than those with CIMP- and CIMP+2 tumors	Among patients treated with surgery alone, those with CIMP+1 tumors reported lower cancer-specific survival and relapse-free survival than those with CIMP- and CIMP+2 tumors

Abbreviations: CRC, Colorectal Cancer; 5-FU, 5 fluorouracil; CIMP, CpG island methylator phenotype; MS-PCR, methylation-specific PCR; CDKN2A, Cyclin Dependent Kinase Inhibitor 2A; MINT-2, Methylated-IN-Tumour locus 2; MDR1, Multi Drug Reactivity 1; CACNA1G, Calcium Voltage-Gated Channel Subunit Alpha1 G; IGF2, Insulin Like Growth Factor 2; NEUROG1, Neurogenin 1; RUNX3, Runt-Related Transcription Factor 3; SOCS1, Suppressor Of Cytokine Signaling 1; MLH1, MutL homolog 1; CRABP1, Cellular Retinoic Acid Binding Protein 1.

**Table 3 ijms-20-03842-t003:** LINE-1 methylation markers in patients receiving adjuvant chemotherapy.

First Author and Year	Tumor	Overall Population (Treated Patients)	Treatment	Samples	Markers	Methods	Response to Treatment	Other Findings
Kawakami, K., 2011 [71]	CRC	155 (94)	5-FU	Tumor tissues	LINE-1	MS-PCR and Methylight assay	LINE-1 hypomethylation conferred a survival benefit in patients receiving adjuvant therapy. Benefits from adjuvant chemotherapy were not evident in patients with LINE-1 hypermethylation	In patients treated with surgery alone, LINE-1 hypomethylation was associated with worse prognosis
Chen, D., 2016 [73]	CRC	336 (NA)	FOLFOX-4 or mFOLFOX-6	Tumor tissues	LINE-1	Pyrosequencing	LINE-1 methylation was associated with clinicopathological features and recurrence-free survival in CRC patients receiving adjuvant therapy based on the FOLFOX protocol	NA
Lou, Y.T., 2015 [74]	Colon cancer	129 (129)	FOLFOX-4	Tumor tissues	LINE-1	Pyrosequencing	LINE-1 methylation level was lower in patients receiving adjuvant therapy with post-therapeutic recurrence than in those without recurrence. LINE-1 hypomethylation as an independent risk factor of post-therapeutic recurrence	Patients with LINE-1 hypomethylation had reduced disease free survival in the whole cohort and in those with post-therapeutic recurrence after 6 months

Abbreviations: CRC, Colorectal Cancer; 5-FU, 5 fluorouracil; LINE-1, Long interspersed nuclear element-1; MS-PCR, methylation-specific PCR.

**Table 4 ijms-20-03842-t004:** Gene-specific methylation markers in rectal cancer patients receiving neoadjuvant chemoradiation.

First Author and Year	Tumor	Overall Population (Treated Patients)	Treatment	Samples	Markers	Methods	Response to Treatment	Other Findings
De Maat, M.F., 2010 [75]	Rectal cancer	251 (251)	Chemoradiation	Tumor tissues	MINT loci	Absolute quantitative assessmen tof methylated alleles	Not examined	MINT3 hypermethylation and MINT17 hypomethylation reduced the risk of recurrence
Ebert, M.P., 2012 [76]	Rectal cancer	220 (NA)	Chemoradiation	Tumor tissues	TFAP2E	MethyLight assay	TFAP2E hypermethylation was associated with non-response to neoadjuvant chemoradiation. The odds of adequate response were higher in patients with TFAP2E hypomethylation compared with the whole cohort	NA
Molinari, C., 2013 [77]	Rectal cancer	74 (74)	Chemoradiation	Tumor tissues	24 genes	Methylation-specific multiplex ligation-dependent probeamplification	In patients receiving neoadjuvant therapy, TIMP3 methylation status was significantly different across four tumor regression grade classes	Compared with adjacent normal tissues, tumor samples exhibited hypermethylation of ESR1, CDH13, RARB, IGSF4, and APC genes
Sun, W., 2013 [78]	Rectal cancer	34 (34)	Chemoradiation	Serum	MGMT	MS-PCR	Serum MGMT hypermethylation was associated with improved response to treatment and higher regression compared with MGMT hypomethylation	NA

Abbreviations: MINT, Methylated-IN-Tumour loci; TFAP2E, Transcription Factor Activating Protein 2 Epsilon; ESR1, Estrogen Receptor 1; CDH13, Cadherin 13; RARB, Retinoic Acid Receptor Beta; IGSF4, Cell Surface Adhesion Molecule; APC, Adenomatous Polyposis Coli; TIMP3, Tissue Inhibitor of Metalloproteinases 3; MS-PCR, methylation-specific PCR; MGMT, O6-methylguanine-DNA methyltransferase.

**Table 5 ijms-20-03842-t005:** CpG island methylator phenotype markers in rectal cancer patients receiving neoadjuvant chemoradiation.

First Author and Year	Tumor	Overall Population (Treated Patients)	Treatment	Samples	Markers	Methods	Response to Treatment	Other Findings
Jo, P., 2012 [79]	Rectal cancer	150 (150)	Chemoradiation	Tumor tissues	CIMP (CACNA1G, IGF2, NEURO1G, RUNX3, and SOCS1)	MS-PCR	No association with response to neoadjuvant chemoradiation was evident	No association of CIMP status with clinicopathological features, and KRAS or BRAF mutations was evident. CIMP+ patients exhibited worse 3-year and 5-year disease-free survival
Kohonen-Corish, M.R., 2013 [80]	Rectal cancer	381 (18)	Chemoradiation	Tumor tissues	CIMP (CACNA1G, IGF2, NEURO1G, RUNX3, and SOCS1) and CDKN2A	MethyLight assay	Not examined	No association of CIMP+ status with overall survival was evident. Combination of CDKN2A methylation and KRAS mutations independently predicted the risk of recurrence, thereby reducing overall and cancer-specific survival

Abbreviations: CIMP, CpG island methylator phenotype; CACNA1G, Calcium Voltage-Gated Channel Subunit Alpha1 G; IGF2, Insulin Like Growth Factor 2; NEUROG1, Neurogenin 1; RUNX3, Runt-Related Transcription Factor 3; SOCS1, Suppressor Of Cytokine Signaling 1; MS-PCR, methylation-specific PCR; CDKN2A, Cyclin Dependent Kinase Inhibitor 2A; KRAS, KRAS Proto-Oncogene.

**Table 6 ijms-20-03842-t006:** Global and genome-wide methylation markers in rectal cancer patients receiving neoadjuvant chemoradiation.

First Author and Year	Tumor	Overall Population (Treated Patients)	Treatment	Samples	Markers	Methods	Response to Treatment	Other Findings
Tsang, J.S., 2014 [81]	Rectal cancer	53 (53)	Chemoradiation	Tumor tissues	Global methylation	Immunohistochemical staining and image analysis of staining	A significant reduction of global DNA methylation in the majority of patients. Global methylation of pre-treatment specimens was significantly correlated with tumor stage and regression	NA
Gaedcke, J., 2014 [82]	Rectal cancer	185 (185)	Chemoradiation	Tumor tissues	Whole genome methylation	Whole genome methylation CpG island array	Not examined	In a discovery cohort, 20 highly discriminative DMRs were identified. In two additional validation cohorts, 10 DMRs that allowed the discrimination of patients with different prognosis were confirmed

**Table 7 ijms-20-03842-t007:** MiRNA signatures in patients receiving adjuvant chemotherapy.

First Author and Year	Tumor	Overall Population (Treated Patients)	Treatment	Samples	Markers	Methods	Response to Treatment	Other Findings
Schetter, A.J., 2008 [83]	Colon cancer	197 (72)	5-FU	Tumor and normal tissues	miR-20a, miR-203, miR-21, miR-106a, miR-181b	Microarray and Real-time PCR	miR-20a, miR-21, miR-106a, miR-181b, and miR-203 were enriched in tumor tissues. MiR-21 upregulation was associated with poor survival and poor therapeutic response in both cohorts	37 miRNAs were differentially expressed between tumor and normal tissues
Oue, N., 2014 [84]	Colon cancer	301 (84)	5-FU	Tumor tissues	miR-21	Real-time PCR	Patients with miR-21 upregulation did not benefit from the adjuvant chemotherapy	Upregulation of miR-21 was associated with poor survival independent of clinicopathological features
Coebergh van den Braak, R.R.J., 2018 [85]	Colon cancer	232 (77)	NA	Tumor and normal tissues	let-7i, miR-10b and miR-30b	Real-time PCR	The 2-miRNA signature (let-7i and miR-10b) had no prognostic value in stage III patients receiving adjuvant therapy	The 2-miRNA signature (let-7i and miR-10b) predicted hepatic recurrence in stage I-II patients, while the combination with miR-30b also predicted distant metastasis
Ma, Y., 2011 [86]	CRC	424 (NA)	NA	Tumor and normal tissues	miR-150	Microarray and Real-time PCR	Low miR-150 expression levels were associated with reduced overall survival and worse response to treatment	MiR-150 was downregulated in tumors compared with adjacent normal tissues
Perez-Carbonell, L., 2015 [87]	CRC	Discovery cohort: 100 (NA) Validation cohort: 237 (167)	FOLFOX, 5-FU, or 5-FU and oxaliplatin	Tumor tissues	miR-320e	Microarray and Real-time PCR	Not examined	MiR-320e was upregulated in patients with recurrence compared with those without recurrence. MiR-320e upregulation was associated with poor overall and disease-free survival and overall survival
Zhang, J.X., 2013 [88]	CRC	Training set: 138 (NA) Internal set: 137 (NA) Validation set: 360 (NA)	5-FU	Tumor tissues	miR-20a-5p, miR-21-5p, miR-103a-3p, miR-106a-5p, miR-143-5p, and miR-215	Microarray and Real-time PCR	Based on a 6-miRNA-based classifier, patients with high risk of disease progression exhibited a better response to 5-FU based adjuvant chemotherapy	35 miRNAs were differentially expressed between tumor and normal tissues. Based on a 6-miRNA-based classifier, patients were classified according to their risk of disease progression. Five-year disease-free survival was higher in low risk patients compare with those at high risk
To, K.K., 2015 [89]	Colon cancer	26 (26)	NA	Tumor tissues	miR-519c	Real-time PCR	Most of CRC samples from non-responders exhibited miR-519c downregulation	MiR-519c levels were positively correlated with ABCG2 expression
Diaz, T., 2014 [90]	CRC	127 (56/24)	5-FU or XELOX	Tumor and normal tissues	miR-200 family, miR-141, miR-429	Real-time PCR	Among patients receiving 5-FU based adjuvant therapy, upregulation of miR-200a, miR-200c, miR-141, or miR-429 was positively associated with overall and disease-free survival	Upregulation of miR-200a, miR-200c and miR-429 was associated with better survival
Li, S., 2013 [91]	CRC	125 (91)	5-FU	Tumor tissues	miR-215	Real-time PCR	MiR-215 downregulation was associated with worse response to 5-FU adjuvant therapy	Patients with recurrence exhibited lower miR-215 expression level than those without recurrence
Dou, R., 2013	CRC	35 (35)	FOLFOX6	Tumor tissues	17 miRNAs	Microarray and Real-time PCR	17 miRNAs were differentially expressed between responders and non-responders	Three genes (AQP9, SATB2, and WIF1) were downregulated by the differentially expressed miRNAs
Conev, N.V., 2015 [93]	Colon cancer	37 (37)	5-FU	Serum	miR-17, miR-21, miR-29a, and miR-92	Real-time PCR	Not examined	MiR-17, miR-21 and miR-92 were upregulated in patients with recurrence
Liu, C., 2016 [94]	Colon cancer	84 (66)	FOLFOX	Serum	miR-4772-3p	RNA-seq and Real-time PCR	Downregulation of miR-4772-3p was associated with an increased risk of recurrence. This evidence was confirmed in patients receiving adjuvant therapy	145 miRNAs were differentially expressed between patients with or without recurrence

Abbreviations: 5-FU, 5 fluorouracil; ABCG2, ATP Binding Cassette Subfamily G Member 2; AQP9, Aquaporin-9; SATB2, Special AT-Rich Sequence-Binding Protein 2; WIF1, Wnt Inhibitory Factor 1.

**Table 8 ijms-20-03842-t008:** MiRNA signatures in rectal cancer patients receiving neoadjuvant chemoradiation.

First Author and Year	Tumor	Overall Population (Treated Patients)	Treatment	Samples	Markers	Methods	Response to Treatment	Other Findings
Gaedcke, J., 2012 [95]	Rectal cancer	57 (57)	Chemoradiation	Tumor tissues	miR-135b	Microarray	Not examined	MiR-492, miR-542-5p, miR-584, miR-483-5p, miR-144, miR-2110, miR-652, miR-375, miR-147b, miR-148a, miR-190, miR-26a/b, and miR-338-3p were differentially expressed between rectal and colon cancer tissues. MiR-135b expression correlated with tumor regression grade, disease-free survival and cancer-specific survival
Drebber, U., 2011 [96]	Rectal cancer	40 (40)	Chemoradiation	Tumor tissues	miR-21, miR-143 and miR-145	Real-time PCR	Compared to pre-treatment tissues, miR-21 was downregulated while miR-143 and miR-145 were upregulated in post-treatment tissues. MiR-145 downregulation was associated with worse response to neoadjuvant therapy	MiR-21 was upregulated in tumor tissue than in adjacent normal mucosa
Svoboda, M., 2012 [97]	Rectal cancer	20 (20)	Chemoradiation	Tumor tissues	miR-215, miR-190b, miR-29b-2, let-7e, miR-196b, miR-450a, miR-450b-5p and miR-99a	Microarray and Real-time PCR	Eight miRNAs were differentially expressed between responders and non-responders. MiR-215, miR-190b and miR-29b-2 were upregulated in non-responders, while let-7e, miR-196b, miR-450a, miR-450b-5p and miR-99a were upregulated in responders. This 8-miRNA signature allowed to correctly classify 90% of non-responders	NA
Kheirelseid, E.A., 2012 [98]	Rectal cancer	12 (12)	Chemoradiation	Tumor tissues	miR-16, miR-590-5p, miR-153, miR-519c-3p, and miR-561	Microarray and Real-time PCR	A 3-miRNA signature (miR-16, miR-590-5p and miR-153) predicted incomplete response to neoadjuvant therapy. A 2-miRNA signature (miR-519c-3p and miR-561) predicted poor response to neoadjuvant therapy with an accuracy of 100%	NA
Della Vittoria Scarpati, G., 2012 [99]	Rectal cancer	38 (38)	Chemoradiation	Tumor tissues	miR-1183, miR-483-5p, miR-622, miR-125a-3p, miR-1224-5p, miR-188-5p, miR-1471, miR-671-5p, miR-1909, miR-630, miR-765, miR-1274b, miR-720	Microarray and Real-time PCR	14 miRNAs were differentially expressed in patients with complete response. 11 miRNAs were upregulated (miR-1183, miR-483-5p, miR-622, miR-125a-3p, miR-1224-5p, miR-188-5p, miR-1471, miR-671-5p, miR-1909, miR-630, miR-765), while 2 miRNAs were downregulated (miR-1274b, miR-720). MiR-622 and miR-630 showed 100% sensitivity and specificity in discriminating patients with complete response	NA
Campayo, M., 2018 [100]	Rectal cancer	96 (96)	Chemoradiation	Tumor tissues	377 miRNAs including miR-21, miR-99b and miR-375	Microarray and Real-time PCR	8 miRNAs (let-7b, let-7e, miR-21, miR-99b, miR-183, miR-328, miR-375 and miR-483-5p) were differentially expressed across different tumor regression grades. The combination of miR-21, miR-99b and miR-375 allowed to discriminate patients with complete response to neoadjuvant therapy	MiR-21, miR-99b and miR-375 were associated with disease-free and overall survival.
Lopes-Ramos, C.M., 2014 [101]	Rectal cancer	Training cohort: 27 (27) Validation cohort: 16 (16)	Chemoradiation	Tumor tissues	miR-21-5p	RNA-seq and Real-time PCR	In the training set, four miRNAs were differentially expressed between complete and incomplete responders. In the validation set, miR-21-5p was upregulated in complete responders. MiR-21-5p expression level allowed the prediction of complete response to neoadjuvant therapy with a sensitivity of 78% and a specificity of 86%	NA
Caramés, C., 2016 [102]	Rectal cancer	78 (78)	Chemoradiation	Tumor tissues	mir-31	Real-time PCR	MiR-31 expression level were higher in patients with complete response than in those with poor response	MiR-31 upregulation was associated with poor pathological response and worse overall survival
D’Angelo, E., 2017 [103]	Rectal cancer	38 (38)	Chemoradiation	Tumor tissues	miR-194	Real-time PCR	MiR-194 was overexpressed in patients who responded to neoadjuvant chemoradiation	NA
Du, B., 2018 [104]	Rectal cancer	38 (38)	Chemoradiation	Tumor tissues	41 miRNAs	Microarray	41 miRNAs were differentially expressed between complete and incomplete responders to neoadjuvant therapy. MiR‑548c‑5p/miR‑548d‑5p and miR‑663a regulated genes associate with rectal cancer, thereby modulating the complete response to neoadjuvant chemoradiation	NA
Luo, J., 2018 [105]	Rectal cancer	55 (55)	Chemoradiation	Tumor tissues	miR-519b-3p	Real-time PCR	In patients receiving neoadjuvant therapy, miR-519b-3p expression was positively correlated with response to treatment. A functional analysis suggested that miR-519b-3p was directly involved in response to neoadjuvant chemoradiation in an ARID4B-dependent way	NA
Yu, J., 2016 [106]	Rectal cancer	149 (149)	Chemoradiation	Tumor tissues and serum	miR-345	Microarray and Real-time PCR	MiR-345 upregulation was associated with a worse response to treatment either in tissue or serum	MiR-345 downregulation in serum was associated with better recurrence-free survival
Menéndez, P., 2013 [107]	Rectal cancer	28 (28)	Chemoradiation	Serum	miR-21	Real-time PCR	Not examined	Serum miR-21 downregulation was associated with high risk of recurrence and death. MiR-21 expression was an independent predictor of overall survival
Hiyoshi, Y., 2017 [108]	Rectal cancer	94 (94)	Chemoradiation	Serum	let-7b, miR-15b, miR-20a, miR-21, miR-29a, miR-92a, miR-122, miR-125b, miR-141, miR-143, miR-145, miR-155, miR-200c, miR-221, miR-345, miR-423, miR-425, miR-1246	Real-time PCR	Serum miR-143 expression was higher in responders than in non-responders. Serum miR-143 expression was an independent predictor of pathological response	NA

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
