# Peer review of "Epigenetic Biomarkers in Colorectal Cancer Patients Receiving Adjuvant or Neoadjuvant Therapy: A Systematic Review of Epidemiological Studies"

_ijms, 2019, doi:10.3390/ijms20153842_

Round 1

Reviewer 1 Report

The paper of Barchitta et al “Epigenetic biomarkers in colorectal cancer patients receiving adjuvant or neoadjuvant therapy: a systematic review of epidemiological studies” is a worthwhile review of the literature on this topic. It is a timely summary on the biomarkers identification in colorectal cancer patients, beside to be instrumental for possible future applications in tumour pharmacology.

The information that can be obtained from the literature is very varied and sometimes dispersive, so it is not easy to draw definitive conclusions. Nevertheless, I believe that some comments should be added to enlarge the discussion on the main findings. I refer in particular to those biomarkers for which information is available in more than one paper and is concordant (for example miR-21 but not only). Well, for these biomarkers the authors could better highlight the relationship with the therapy specifically administered. 

Also some comments are worthwhile and can be added on the nature of the sample analysed for biomarker expression (tissue vs serum).

Other points:

·      Lines 112-113. The sum of the studies is 54 and not 52.

·      There are no references on Tables 2, 3, 4, 5, and 6 in the text.

·      Abbreviations should be defined.

·      Table 5 reports the same content of Table 4.

·      Table 6 is erroneously marked as Table 4.

Lines 431-433. The information reported does not sound and in fact it is incorrect: miR-31 level is higher in patients with poor response.

Author Response

Dear Editor,

Thank you very much for considering our manuscript and for suggestions of independent Reviewers. We submit to your attention a revised version of the manuscript in which we have considered all comments. The following List of change and answers to comments of Reviewers addresses all changes made in the manuscript (red font).

The paper of Barchitta et al “Epigenetic biomarkers in colorectal cancer patients receiving adjuvant or neoadjuvant therapy: a systematic review of epidemiological studies” is a worthwhile review of the literature on this topic. It is a timely summary on the biomarkers identification in colorectal cancer patients, beside to be instrumental for possible future applications in tumour pharmacology.

We are very grateful with Reviewer 1 for his/her positive comments on our manuscript.

The information that can be obtained from the literature is very varied and sometimes dispersive, so it is not easy to draw definitive conclusions. Nevertheless, I believe that some comments should be added to enlarge the discussion on the main findings. I refer in particular to those biomarkers for which information is available in more than one paper and is concordant (for example miR-21 but not only). Well, for these biomarkers the authors could better highlight the relationship with the therapy specifically administered. Also some comments are worthwhile and can be added on the nature of the sample analysed for biomarker expression (tissue vs serum).

We agree with Reviewer 1 that the discussion section should provide more details and definitive conclusions of our findings. Accordingly, we revised all the discussion section by including some paragraphs on specific biomarkers (e.g. MGMT methylation, CIMT and miR-21) with robust findings obtained from tissue and blood samples.

Other points:

Lines 112-113. The sum of the studies is 54 and not 52.

We apologize for this mistake that has been corrected in the revised version of our manuscript.

There are no references on Tables 2, 3, 4, 5, and 6 in the text.

According to Reviewer 1 comment, we added references to table in lines 145 and 295.

Abbreviations should be defined.

As requested, we defined all the abbreviations throughout the text and in Table footnotes

Table 5 reports the same content of Table 4.

Table 6 is erroneously marked as Table 4.

We apologize for these mistakes that have been corrected in the revised version of our manuscript.

Lines 431-433. The information reported does not sound and in fact it is incorrect: miR-31 level is higher in patients with poor response.

We are very grateful with Reviewer 1 for this/her careful revision that help us in improving our manuscript. Accordingly, we modified the information reported in line 471.

Reviewer 2 Report

The Review from Barchitta M et al entitled “Epigenetic biomarkers in colorectal cancer patients receiving adjuvant or neoadjuvant therapy: a systematic review of epidemiological studies” investigates epigenetic biomarkers in colorectal patients receiving neoadjuvant and adjuvant therapy, and their potential role for the prediction of outcomes and response to treatment. Although their work is potentially interesting, there are some points that need to be expanded better.

1)    The authors should insert a short paragraph describing epigenetic mechanisms (DNA methylation, Histone modifications and microRNA).

2)    Table entitled “LINE-1 methylation markers in rectal cancer patients receiving neoadjuvant chemoradiation” should be corrected in Table 6.

3)    In Tables 4, 5 and 6 the authors should insert two new columns: one entitled “Tumor” (between the first and the second column) and the other entitled  “Treatment” (before the second and the third column). This is requested for all the tables to be the same.

4)     I would recommend authors to insert new tables to compare epigenetic markers in CRC patients receiving adjuvant ad neoadjuvant therapy (i.e. a table to compare gene-specific methylation markers or LINE-1 methylation markers in CRC patients receiving adjuvant ad neoadjuvant therapy)

5)    In all tables, please provide also First author’s initial name .

Author Response

Dear Editor,

Thank you very much for considering our manuscript and for suggestions of independent Reviewers. We submit to your attention a revised version of the manuscript in which we have considered all comments. The following List of change and answers to comments of Reviewers addresses all changes made in the manuscript (red font).

The Review from Barchitta M et al entitled “Epigenetic biomarkers in colorectal cancer patients receiving adjuvant or neoadjuvant therapy: a systematic review of epidemiological studies” investigates epigenetic biomarkers in colorectal patients receiving neoadjuvant and adjuvant therapy, and their potential role for the prediction of outcomes and response to treatment. Although their work is potentially interesting, there are some points that need to be expanded better.

We are very grateful with Reviewer 2 for his/her positive comments on our manuscript. 

1)    The authors should insert a short paragraph describing epigenetic mechanisms (DNA methylation, Histone modifications and microRNA).

As requested, we added brief paragraphs on DNA methylation and miRNAs in the introduction section. With regard to histone modification, we prefer not to address the topic since we found no articles on this kind of epigenetic mechanism. However, some references on histone modifications are provided in the discussion section.

2)    Table entitled “LINE-1 methylation markers in rectal cancer patients receiving neoadjuvant chemoradiation” should be corrected in Table 6.

We apologize for this mistake that has been corrected in the revised version of our manuscript.

3)    In Tables 4, 5 and 6 the authors should insert two new columns: one entitled “Tumor” (between the first and the second column) and the other entitled  “Treatment” (before the second and the third column). This is requested for all the tables to be the same.

According to Reviewer 2 suggestion, we added two columns for “Tumor” and “Treatment” in each table where they were missing.

4)     I would recommend authors to insert new tables to compare epigenetic markers in CRC patients receiving adjuvant ad neoadjuvant therapy (i.e. a table to compare gene-specific methylation markers or LINE-1 methylation markers in CRC patients receiving adjuvant ad neoadjuvant therapy)

We really appreciate this suggestion by Reviewer 2, but we prefer not to add additional tables since results obtained in patients receiving adjuvant or neoadjuvant therapies are very heterogenous. However, we reported some comparisons in the discussion section for those biomarkers that were deeply investigated. 

5)    In all tables, please provide also First author’s initial name.

As suggested, we included First author’s initial name in all tables.

Reviewer 3 Report

The authors investigated in their systematic review the potential influence of epigenetic biomarkers to predict response and outcome of patients with colon and rectal cancer focusing on DNA methylation and expression of microRNAs. As described by the authors it is important in the future in order to further individualize and target adjuvant and neoadjuvant treatment to identify subgroups which benefit from treatment or are even harmed. Therefore, much more information about the value of biomarkers is needed for these entities.

Here, the authors provide in their review one piece of information that might help researchers in the field to get an quick overview about this topic and to get inspired to continue to finally help to sub-classify and individualize treatment of colon and rectal cancer.

The manuscript is well written and readily understandable. The text of the results section perfectly matches the tables. The tables nicely summarize the studies described in the text.

Minor points:

Line 465 Discussion about toxicity: The authors should mention the problem of neurotoxicity of oxaliplatin. This is a frequent, sometimes persisting chronic problem for many patients.

Line 474-475: several panels..... has been. This should be have been.  

The manuscript also includes several double spaces in the text that need to be removed. 

In summary, this manuscript is well written and nicely summarizes the examined field of the association of epigenetic biomarkers and adjuvant and neoadjuvant therapy in colon and rectal cancer in an open and unbiased way.

Author Response

Dear Editor,

Thank you very much for considering our manuscript and for suggestions of independent Reviewers. We submit to your attention a revised version of the manuscript in which we have considered all comments. The following List of change and answers to comments of Reviewers addresses all changes made in the manuscript (red font).

The authors investigated in their systematic review the potential influence of epigenetic biomarkers to predict response and outcome of patients with colon and rectal cancer focusing on DNA methylation and expression of microRNAs. As described by the authors it is important in the future in order to further individualize and target adjuvant and neoadjuvant treatment to identify subgroups which benefit from treatment or are even harmed. Therefore, much more information about the value of biomarkers is needed for these entities. Here, the authors provide in their review one piece of information that might help researchers in the field to get an quick overview about this topic and to get inspired to continue to finally help to sub-classify and individualize treatment of colon and rectal cancer. The manuscript is well written and readily understandable. The text of the results section perfectly matches the tables. The tables nicely summarize the studies described in the text.

We are very grateful with Reviewer 2 for his/her positive comments on our manuscript. 

Minor points:

Line 465 Discussion about toxicity: The authors should mention the problem of neurotoxicity of oxaliplatin. This is a frequent, sometimes persisting chronic problem for many patients.

As suggested by Reviewer 3, we added additional comment on the toxicity of oxaliplatin in the discussion section.

Line 474-475: several panels..... has been. This should be have been.  

The manuscript also includes several double spaces in the text that need to be removed. 

We apologize for these mistakes that have been corrected in the revised version of our manuscript.

In summary, this manuscript is well written and nicely summarizes the examined field of the association of epigenetic biomarkers and adjuvant and neoadjuvant therapy in colon and rectal cancer in an open and unbiased way.

Round 2

Reviewer 2 Report

The authors are requested to insert also a brief description of histone modifications in general.

Author Response

Dear Editor,

Thank you very much for considering our manuscript and for suggestions of independent Reviewers. We submit to your attention a revised version of the manuscript in which we have considered all comments. The following List of change and answers to comments of Reviewers addresses all changes made in the manuscript (blue font).

Reviewer 2

The authors are requested to insert also a brief description of histone modifications in general.

As requested by Reviewer 2, we included a brief description of histone modifications.

Round 3

Reviewer 2 Report

Accept in present form